# Antibiotic Resistance Patterns of Bacterial Isolates from Neonatal Sepsis Patients at University Hospital of Leipzig, Germany

**DOI:** 10.3390/antibiotics10030323

**Published:** 2021-03-19

**Authors:** Belay Tessema, Norman Lippmann, Matthias Knüpfer, Ulrich Sack, Brigitte König

**Affiliations:** 1Institute of Medical Microbiology and Epidemiology of Infectious Diseases, Faculty of Medicine, University of Leipzig, 04103 Leipzig, Germany; Norman.Lippmann@medizin.uni-leipzig.de (N.L.); Brigitte.Koenig@medizin.uni-leipzig.de (B.K.); 2Institute of Clinical Immunology, Faculty of Medicine, University of Leipzig, 04103 Leipzig, Germany; Ulrich.Sack@medizin.uni-leipzig.de; 3Department of Medical Microbiology, College of Medicine and Health Sciences, University of Gondar, Gondar 196, Ethiopia; 4Department of Neonatology, Faculty of Medicine, University of Leipzig, 04103 Leipzig, Germany; Matthias.Knuepfer@medizin.uni-leipzig.de

**Keywords:** antibiotic resistance, bacterial pathogens, neonatal sepsis

## Abstract

Neonatal sepsis caused by resistant bacteria is a worldwide concern due to the associated high mortality and increased hospitals costs. Bacterial pathogens causing neonatal sepsis and their antibiotic resistance patterns vary among hospital settings and at different points in time. This study aimed to determine the antibiotic resistance patterns of pathogens causing neonatal sepsis and to assess trends in antibiotic resistance. The study was conducted among neonates with culture proven sepsis at the University Hospital of Leipzig between November 2012 and September 2020. Blood culture was performed by BacT/ALERT 3D system. Antimicrobial susceptibility testing was done with broth microdilution method based on ISO 20776-1 guideline. Data were analyzed by SPSS version 20 software. From 134 isolates, 99 (74%) were gram positive bacteria. The most common gram positive and gram negative bacteria were *S. epidermidis*, 51 (38%) and *E. coli*, 23 (17%), respectively. *S. epidermidis* showed the highest resistance to penicillin G and roxithromycin (90% each) followed by cefotaxime, cefuroxime, imipenem, oxacillin, and piperacillin-tazobactam (88% each), ampicillin-sulbactam (87%), meropenem (86%), and gentamicin (59%). Moreover, *S. epidermidis* showed raising levels of resistance to amikacin, gentamicin, ciprofloxacin, levofloxacin, moxifloxacin, and cotrimoxazol. Gram positive bacteria showed less or no resistance to daptomycin, linezolid, teicoplanin, and vancomycin. *E. coli* showed the highest resistance to ampicillin (74%) followed by ampicillin-sulbactam (52%) and piperacillin (48%). Furthermore, increasing levels in resistance to ampicillin, ampicillin-sulbactam, piperacillin, and cefuroxime were observed over the years. Encouragingly, *E. coli* showed significantly declining trends of resistance to ciprofloxacin and levofloxacin, and no resistance to amikacin, colistin, fosfomycin, gentamicin, imipenem, piperacillin-tazobactam, and tobramycin. In conclusion, this study demonstrates that gram positive bacteria were the leading causes of neonatal sepsis. Bacterial isolates were highly resistant to first and second-line empiric antibiotics used in this hospital. The high levels of antibiotic resistance patterns highlight the need for modifying empiric treatment regimens considering the most effective antibiotics. Periodic surveillance in hospital settings to monitor changes in pathogens, and antibiotic resistance patterns is crucial in order to implement optimal prevention and treatment strategies.

## 1. Introduction

Neonatal sepsis is a clinical syndrome in neonates, manifesting with a non-specific systemic signs and symptoms due to bloodstream infection [1]. Hospitalized neonates, especially premature neonates, are vulnerable to nosocomial infections due to immaturity of the immune system and the need for several invasive procedures [2,3,4,5,6]. An estimated 2.5 million neonatal deaths occur each year, representing 47% of all deaths in children younger than 5 years globally [7]. Bacterial infections are the leading cause of global neonatal deaths, and the risk of mortality from neonatal sepsis is higher than the risk of mortality from other neonatal conditions [8].

Commencing early antibiotic treatment is essential for successful prognosis of neonatal sepsis patients. So, treatment is frequently started before getting blood culture results [9,10,11,12,13,14]. Many physicians also decide empirical treatment for presumed neonatal sepsis due to the problem of collecting adequate blood samples for culture, the long time required to get culture results, and the low sensitivity of blood culture method [9,11]. World Health Organization guidelines for the management of suspected neonatal infections recommend empirical treatment with ampicillin combined with gentamicin as first line therapy, with a third-generation cephalosporin as second-line therapy for non-responders or patients in whom drug-susceptibility testing of bacterial isolates indicates resistance to first-line therapy [15]. Other authors have also recommended empirical treatment of early onset sepsis (EOS) with ampicillin combined with gentamicin as first line therapy [13,16,17]. For empirical treatment of late onset sepsis (LOS), variable recommendations have been made by different authors. These include ampicillin combined with gentamicin [16,18], vancomicin combined with gentamicin for nosocomial LOS [16], and piperacillin-tazobactam for both EOS and LOS [19]. However, empirical therapy is often inappropriate, with unnecessary broad-spectrum antibiotics use and a prolonged duration of treatment contributing to an increasing number of drug-resistant microorganisms [18,20].

Unnecessary and over use of antibiotics, particularly broad-spectrum antibiotics, is already documented as a significant factor for the emergence of drug resistant strains [2,9,10,12,14]. Recently, the rise of antibiotic resistant strains has been reported by several studies in patient populations [9,10]. The rise of neonatal sepsis caused by antibiotic resistant bacteria is a multi-factorial problem. Now a days, it is a global concern because of the associated high illness and death, as well as increased hospitals costs for the management of patients [2,11].

Bacterial pathogens responsible for neonatal sepsis and their antibiotic resistance patterns vary with the geographical areas and at different points in time [21] depending upon the prevalent local pathogens and common antibiotic used in the neonatal department [22]. The first-line antibiotic regimen recommended for empirical therapy should be modified to address the most common local pathogens and their antibiotic resistance patterns. Hence, the bacterial pathogens responsible and their antibiotic resistance patterns should be regularly monitored in a hospital setting in order to select appropriate antibiotic therapy to decrease neonatal mortality. Therefore, this study is conducted to determine the antibiotic resistance pattern of pathogens causing neonatal sepsis, and to assess the trends in antibiotic resistance of common bacterial pathogens in the University Hospital of Leipzig, thereby to provide antibiogram to neonatologists for better management of neonatal sepsis.

## 2. Results

### 2.1. Characteristics of Study Participants and Organisms Isolated from Sepsis Patients

A total of 152 neonates with culture positive results were registered in the laboratory record during the study period. However, 18 neonates, positive for coagulase negative staphylococci (CoNS) organisms were omitted from the final analysis because of suspicion of contamination as the concentration of C-reactive protein (CRP) was low, <10 mg/L. Consequently, 134 neonates with culture proven sepsis, caused by 19 different types of organisms and tested for antimicrobial resistance were included in this study. Higher proportion, 74 (55.2%) of the newborns were male. The mean ± standard deviation (SD) age of newborns was 11.91 (9.07) days, and 116 (86.6%) of newborns were in the age group of >72 h, late onset sepsis (LOS) cases. Majority of the isolates, 99 (73.9%) were gram positive bacteria. The most predominant isolates were *S. epidermidis*, 51 (38.1%) followed by *E. coli*, 23 (17.2%); *S. haemolyticus*, 15 (11.2); and *S. aureus*, 11 (8.2%). *E. coli* was the predominant pathogen of early onset sepsis (EOS), five (3.7%) while *S. epidermidis* was the predominant pathogen of LOS cases, 49 (36.6%) (Table 1).

### 2.2. Antibiotic Resistance Patterns of Gram Positive Bacteria

The antibiotic resistance patterns of predominant gram positive bacteria between 2012 and 2020 are presented in Table 2. As expected, *S. epidermidis* showed a high level of resistance against all beta lactam antibiotics, fluoroquinolones, and aminoglycocides tested. Luckily, no resistance was observed to daptomycin, linezolid, and glycopeptides (teicoplanin and vancomycin). *S. haemolyticus* showed the highest resistance to ceftaroline (100%); roxithromycin (93.3%); cefotaxime, cefuroxime, ciprofloxacin, imipenem, oxacillin, penicillin G, piperacillin-tazobactam (86.7% each). Less resistance was observed to daptomycin (7.1%); doxycycline, linezolid (6.7% each), and no resistance was observed to vancomycin. *S. aureus* showed highest resistance to penicillin G (72.7%) followed by roxithromycin (27.3%). However, no resistance was observed to most of the other antibiotics tested. *S. agalactiae* showed highest resistance to doxycycline (83.3%) followed by fosfomycin (40%) but no resistance was observed to most of the other antibiotics tested.

Trends in antibiotic resistance of *S. epidermidis* between 2017 and 2020 compared with the period from 2013 to 2016 are summarized in Table 3. *S. epidermidis* showed increasing levels of resistance against amikacin, gentamicin, ciprofloxacin, levofloxacin, moxifloxacin, and cotrimoxazol. Interestingly, it revealed decreasing levels of resistance against roxithromycin, oxacillin, cefotaxime, cefuroxime, clindamycin, imipenem, ampicillin-sulbactam, meropenem, and piperacillin-tazobactam. Moreover, *S. epidermidis* showed no changes in the levels of resistance against all other drugs tested between 2017 and 2020 compared with the level of resistance between 2013 and 2016. More importantly, this bacteria did not develop resistance to daptomycin, linezolid, teicoplanin, and vancomycin from 2013 to 2020, The increase and decrease levels of resistance against the aforementioned antibiotics by S. epidermidis were with no statistically significant changes.

### 2.3. Antibiotic Resistance Patterns of Gram Negative Bacteria

The antibiotic resistance patterns of predominant gram negative bacteria are presented in Table 4. *E. coli* showed the highest resistance to ampicillin (73.9%) followed by ampicillin-sulbactam (52.2%) and piperacillin (47.8%). Moreover, it showed the highest susceptible, increased exposure to cefuroxime (69.6%) followed by piperacillin (26.1%) and ampicillin-sulbactam (21.7%). No resistance was observed to amikacin, colistin, fosfomycin, gentamicin, imipenem, piperacillin-tazobactam, and tobramycin. Of the three tested *Klebsiella oxytoca* isolates, three were resistant to ampicillin, two resistant to fosfomycin, one resistant to piperacillin and two isolates were susceptible, increased exposure to cefuroxime. No resistance was observed to other antibiotics tested. Of the two *Klebsiella pneumonia* isolates, two were resistant to ampicillin, one to ampicillin-sulbactam and piperacillin, and two isolates were susceptible, increased exposure to cefuroxime. No resistance was observed to other antibiotics tested. Enterobacter cloacae (*n* = 2) showed resistance to ampicillin, ampicillin-sulbactam and cefuroxime (2/2 each); ceftibuten and fosfomycin (1/2 each) and susceptible, increased exposure to tobramycin (1/2). No resistance was observed to other antibiotics tested.

Trends in antibiotic resistance of *E. coli* between 2017 and 2020 compared with the period from 2013 to 2016 are summarized in Table 5. *E. coli* showed raising trends of resistance against ampicillin, ampicillin-sulbactam, piperacillin, and cefuroxime. *E.coli* revealed declining trends of resistance against ciprofloxacin, levofloxacin, moxifloxacin, cefotaxime, ceftazidime, aztreonam, and cotrimoxazol. *E. coli* also showed no changes in the level of resistances against all other drugs tested from 2017 to 2020 compared with the level of resistance from 2013 to 2016. The declining trends of resistance to ciprofloxacin and levofloxacin were statistically significant (*p* = 0.030).

## 3. Discussion

In our study, the majority of bacterial isolates from neonatal sepsis patients were CoNS bacteria. Moreover, *S. epidermidis* were the predominant pathogen of LOS while *E. coli* was the leading cause of EOS. These findings are in agreement with the findings of the previous reports [23,24,25,26]. The high risk of developing neonatal LOS caused by CoNS might be due to the immaturity of their immune system, and neonates might undergo invasive procedures and resuscitation which are predisposing them for possible invasive colonization with CoNS bacteria which are normally found on the skin of the neonates as in the case of *S. epidermidis*. The predominance of *E. coli* isolates in EOS cases could also be due to the fact that newborns most probably acquire these gram negative bacteria from the vaginal and fecal flora of the mother and the environment where the delivery occurs [21]. Previous studies have reported that majority of the *E. coli* strains isolated from neonatal sepsis patients possess the K1 capsular polysaccharide antigen as an essential virulence factor, and most of the neonatal sepsis causing *E. coli* strains, 59–70% were K1 antigen positive [27,28,29].

In this study, we found higher proportion of positive blood cultures among LOS cases than in EOS cases. Similar findings were reported in other studies from Egypt and South Africa [30,31]. The conflicting results were also documented in studies from Nepal [32] and Iran [33]. The lower proportion of bacterial isolates in EOS cases might be due to the use of antibiotics during obstetric care. The use of antibiotics for obstetric care might influence the blood culture results of the newborns as there is a substantial transplacental transfer of antibiotics to the fetus.

*S. epidermidis*, the predominant gram positive bacteria, showed high level resistance to ampicillin-sulbactam, cefotaxime, imipenem, and gentamicin. *E. coli,* the predominant gram negative bacteria, also showed high level resistance to ampicillin and ampicillin-sulbactam. Fortunately, *E. coli* showed no resistance to gentamicin and less resistance to cefotaxime in this study. Similarly, other previous studies showed high resistance rates of isolated gram positive and gram negative bacteria against first line antibiotics, ampicillin and gentamicin [17,34].

In addition to the first-line and second-line empirical antibiotic regimens, *S. epidermidis* showed the highest level of resistance to penicillin G, roxithromycin, and cefuroxime during the study period. *S. epidermidis* and *S. hemolyticus* also showed high level of resistance against fifth generation cephalosporin (ceftaroline). The high level resistance rates of CoNS against the third and fifth generation cephalosporin antibiotics is worrisome and may lead to the spread of antibiotic resistant infections especially in hospital settings. *S. epidermidis* also showed increasing levels of resistance against amikacin, gentamicin, ciprofloxacin, levofloxacin, moxifloxacin, and cotrimoxazol over the years. However, these increasing trends of resistance rates were not statistically significant. The high resistance rates of CoNS bacteria and changes in the level of resistance against these antibiotics over the years observed in our study is consistent with the findings of the previous studies [35,36,37,38].

The findings of the current study revealed very encouraging results in that gram-positive bacteria are susceptible to daptomycin, doxycycline, linezolid, teicoplanin, and vancomycin, which is also supported by the findings of other studies [23,24,25,36]. Daptomycin and doxycycline are contra-indicated in neonatal patients, however, the other three antibiotics, vancomycin, teicoplanin, and linezolid could be considered as safe antibiotics of choice for the successful empiric treatment of suspected neonatal sepsis cases caused by gram positive bacterial infections.

*E. coli*, the predominant gram negative bacteria, had the highest overall proportion of resistance to ampicillin followed by ampicillin-sulbactam and piperacillin during the study period. Moreover, *E. coli* showed raising trends of resistance against ampicillin, ampicillin-sulbactam, piperacillin, and cefuroxime during the study period. However, it is very promising that *E. coli* showed significantly declining trends of resistance to ciprofloxacin and levofloxacin over the years. Moreover, all *E. coli* isolates were susceptible to gentamicin, amikacin, colistin, fosfomycin, imipenem, piperacillin-tazobactam, and tobramycin. Similar findings were reported by previous studies [39,40]. Most of these effective drugs could be considered as antibiotic of choice for empiric treatment of suspected neonatal sepsis caused by gram negative bacteria in the future. Nevertheless, we do not advice to generally replace primary antibiotic regimens by third class cephalosporines to avoid the potential emergence of resistant strains afterwards.

Unfortunately, in case of EOS patients, commonly identified gram negative bacteria, *E.coli* was highly resistant to the first line empiric antibiotics. Despite this alarming result, the finding was carefully discussed by neonatologists of this hospital and agreed to continue using first line antibiotic treatment with ampicillin and gentamycin. However, it was decided to rapidly add cefotaxim in the treatment regimen if the newborns with EOS do not get a clinically stable state in a short time. This decision was made in accordant with the recommendation of recently published review [41]. The use of cefotaxime regularly in first empiric therapy is not advisable. Because previous studies have shown that in first line therapy it results in more Extended-spectrum beta lactamase bacteria infections [42] and leads to more invasive fungal infections [43]. Furthermore, Clark and his colleagues [44] found a higher mortality in cefotaxime/ampicillin treated newborns compared with ampicillin/gentamycin-treated newborns.

In general, the high antibiotic resistance rates among gram positive and gram negative bacterial isolates with increasing and decreasing trends during the study period might be due to one or more of the following reasons: The emergence of antimicrobial resistance is a normal evolutionary process for microorganisms which is accelerated by the selective pressure exerted by widespread use and misuse of antibiotics; poor infection and disease prevention and control in health-care facilities; poor access to quality and affordable medicines, vaccines, and diagnostics; lack of awareness and knowledge; and lack of enforcement of legislation that could accelerate the emergence and spread of antibiotic resistance [45].

Neonatal sepsis, a life-threatening condition, needs immediate empirical antibiotic therapy. Empirical antibiotic regimens should be guided by the local antibiotic resistance patterns of bacterial isolates commonly detected in the hospital or in the community settings [46]. In our hospital, the first line regimen for empirical treatment of EOS was ampicillin combined with cefotaxime until 2018. Since 2018, this regimen has been changed into ampicillin combined with gentamicin, and in the absence of clinical improvement or in case of suspicion of meningitis cefotaxime is added. For LOS, first line therapy consists of cefotaxime and vancomycin until the blood culture results are available. If there is an abdominal focus, empirical treatment is changed to imipenem. International guidelines on neonatal sepsis management also recommend ampicillin combined with gentamycin as first-line empiric therapy [15,17].

This study had some limitations. First, this retrospective study is based on the data collected form laboratory records which lack information about the neonates’ hospitalization date, clinical information, and treatment outcome. Therefore, we were not able to classify infections as community acquired or hospital acquired infection. Similarly, we could not determine whether the antibiotic resistance was primary or secondary resistance. Moreover, data on the clinical information and treatment outcome of the neonates were not included in this study. Second, this study was conducted only at a single hospital; therefore, the antibiotic resistance patterns observed in our study might not generalize the situation in the country, even though other reports in the country supported our findings.

## 4. Materials and Methods

### 4.1. Study Design and Period

A retrospective cross-sectional study design was used, and data were collected among newborns with proven sepsis diagnosed at the University Hospital of Leipzig, Germany between November 2012 and September 2020. The basic information about newborns such as gender, age, organisms isolated and their antimicrobial susceptibility test results were collected from the laboratory records of the Institute of Medical Microbiology and Epidemiology of Infectious Diseases.

### 4.2. Study Participants

Newborns with positive blood culture results, diagnosed during the study period and with antibiotic susceptibility test results were participated in this study. Newborns with positive blood cultures for CoNS organisms and low CRP concentrations, <10 mg/L were considered as potential contamination and omitted from our final analysis. Newborns were grouped as EOS cases when sepsis onset is in ≤72 h of newborns after birth, or LOS cases when sepsis onset is >72 h of newborns after birth [47].

### 4.3. Blood Culture and Identification of Organisms

From sepsis-suspected neonate, blood sample was collected and cultured using automated BacT/ALERT 3D system, a positive culture was sub-cultured, as described in our previously published work [47]. Pure colonies of bacteria or yeasts isolated from the blood culture were characterized to the species level with Vitek matrix-assisted laser desorption ionization time-of-flight (MALDI-TOF) mass spectrometry (BioMérieux, Marcy L’Etoile, France).

### 4.4. Antimicrobial Susceptibility Testing (AST)

Minimum inhibitory concentrations (MIC) were determined using broth microdilution method according to ISO 20776-1 standard procedures. Isolates suspended to a density of McFaland standard 0.5 were mixed with equal volumes of 9 serial dilutions for each antibiotics (0.03125 to 512 mg/L) in microtiter plates and adjusted to cover the range of susceptible (S), susceptible, increased exposure (I) and resistant (R) results and incubated at 37 °C for 24 h. The MIC results were read using TECAN Sunrise reader with multi-channel network (MCN)6 software program (MERLIN Diagnostika GmbH, Berlin, Germany). Antibiotic susceptibilities were interpreted as susceptible, resistant or susceptible, increased exposure based on the clinical breakpoints established by the European committee on antimicrobial susceptibility testing in 2019 (Version 9.0). For those organisms without defined breakpoint, only the MIC values were reported. The MIC is the lowest concentration of the antibiotic that inhibits the bacterial growth.

### 4.5. Statistical Analyses

Data were first entered into excel sheet. Then data were transferred and analyzed using Statistical Package for the Social Sciences (SPSS, Chicago, IL, USA) version 20 software. The normality of the data distribution was tested by Skewness and Kurtosis Z-values, and the Shapiro–Wilk test *p*-value. Moreover, normality was checked using visual outputs including histograms, normal Q–Q plots, and box plots. Non-parametrical tests were used when the data were not normally distributed. Descriptive statistics; frequency and percentages of organisms isolated from culture positive newborns were calculated based on the onset of sepsis and gender of the newborns. The mean and SD were calculated to measure the mean age of newborns. The Chi-square test (Cochran–Armitage) for linear trend was used to test the significance of annual trends in antimicrobial resistance. For the trend analysis we had to categorize all the isolates either under S or R strain categories. As I (susceptible, increased exposure) isolates require increased dose of the antibiotics for successful treatment, such isolates were categorized under resistant strain category only for the trend analysis tables. A *p*-value of < 0.05 was considered statistically significant.

## 5. Conclusions

This study demonstrates that CoNS bacteria were the leading causes of neonatal sepsis in the study area. *E. coli* was the predominant pathogen of EOS while *S. epidermidis* was the predominant pathogen of LOS cases. Bacterial isolates were highly resistant to first-line and second-line empiric antibiotics used at this hospital for the management of neonatal sepsis. Fortunately, gram positive bacterial isolates showed less or no resistance to daptomycin, linezolid, rifampicin, teicoplanin, and vancomycin. Similarly, gram negative bacterial isolates were sensitive to amikacin, colistin, gentamicin, imipenem, meropenem, piperacillin-tazobactam, and tobramycin. The high levels of resistance to first and second-line empirical antibiotic regimens highlight the need for modifying the treatment regimens considering the most effective antibiotics observed in this study. We also recommend periodic surveillance at hospital settings to monitor changes in pathogens causing neonatal sepsis, and their antibiotic resistance patterns in order to implement optimal prevention and treatment strategies. Moreover, antibiotic rotation through systematically rotating antibiotics or antibiotic classes for empirical treatment might also be helpful to reduce antibiotic resistance. Further studies should be done to compare antibiotic resistance in hospital-acquired-infections and community-acquired infections among neonatal sepsis patients to devise targeted and effective interventions.

## Figures and Tables

**Table 1 antibiotics-10-00323-t001:** Frequency of organisms isolated from neonatal sepsis patients according to sepsis onset, gender, and age of the neonates.

Organism	Total N (%)	Sepsis Onset		Gender		Age in Days
		EOS N (%)	LOS N (%)	Male N (%)	Female N (%)	Mean (± SD)
Gram positive	99 (73.9)	13 (9.7)	86 (64.2)	55 (41.1)	44 (32.8)	11.46 (8.46)
*Staphylococcus epidermidis*	51 (38.1)	2 (1.5)	49 (36.6)	28 (20.9)	23 (17.2)	11.57 (8.41)
*Staphylococcus haemolyticus*	15 (11.2)	1 (0.7)	14 (10.4)	11 (8.2)	4 (3.0)	15.27 (8.17)
*Staphylococcus aureus*	11 (8.2)	-	11 (8.2)	5 (3.7)	6 (4.5)	10.55 (5.41)
*Streptococcus agalactiae*	6 (4.5)	2 (1.5)	4 (3.0)	3 (2.2)	3 (2.2)	12.83 (12.51)
*Staphylococcus hominis*	5 (3.7)	3 (2.2)	2 (1.5	2 (2.7)	3 (2.2)	6.80 (9.39)
*Enterococcus faecalis*	5 (3.7)	-	5 (3.7)	2 (1.5)	3 (2.2)	11.00 (6.78)
*Bacillus cereus*	2 (1.5)	2 (1.5)	-	1 (0.7)	1 (0.7)	2.00
*Listeria monocytogenes*	2 (1.5)	2 (1.5)	-	1 (0.7)	1 (0.7)	0.50 (0.71)
*Micrococcus luteus*	1 (0.7)	-	1 (0.7)	1 (0.7)	-	19.00
*Staphylococcus lugdunensis*	1 (0.7)	1 (0.7)	-	1 (0.7)	-	1.00
Gram negative	34 (25.4)	5 (3.7)	29 (21.7)	18 (13.4)	16 (12.0)	13.27 (10.86)
*Escherichia coli*	23 (17.2)	5 (3.7)	18 (13.4)	15 (65.2)	8 (34.8)	11.83 (10.84)
*Klebsiella oxytoca*	3 (2.2)	-	3 (2.2)	1 (0.7)	2 (2.2)	24.33 (5.86)
*Klebsiella pneumoniae*	2 (1.5)	-	2 (1.5)	-	2 (2.2)	10.50 (6.36)
*Enterobacter cloacae*	2 (1.5)	-	2 (1.5)	1 (0.7)	1 (0.7)	20.00 (9.90)
*Enterobacter hormaechei*	1 (0.7)	-	1 (0.7)	-	1 (0.7)	4.00
*Citrobacter freundii*	1 (0.7)	-	1 (0.7)	-	1 (0.7)	32.00
*Morganella morganii*	1 (0.7)	-	1 (0.7)	-	1 (0.7)	11.00
*Pseudomonas aeruginosa*	1 (0.7)	-	1 (0.7)	1 (0.7)	-	5.00
Fungi	1 (0.7)	-	1 (0.7)	1 (0.7)	-	12.00
*Candida albicans*	1 (0.7)	-	1 (0.7)	1 (0.7)	-	12.00
Total	134 (100)	18 (13.4)	116 (86.6)	74 (55.2)	60 (44.8)	11.91 (9.07)

N = number, EOS = Early onset sepsis, LOS = Late onset sepsis, SD = Standard deviation.

**Table 2 antibiotics-10-00323-t002:** Antibiotic resistance patterns of predominant gram positive bacterial isolates from neonatal sepsis patients, 2012–2020.

Antibiotics	*Staphylococcus epidermidis*		*Staphylococcus haemolyticus*		*Staphylococcus aureus*	*Streptococcus agalactiae*
	R/N (%)	I/N (%)	R/N (%)	I/N (%)	R/N (%)	R/N (%)
Penicillin G	46/51 (90.2)	-	13/15 (86.7)	-	8/11 (72.7)	0/6
Roxithromycin	46/51 (90.2)	-	14/15 (93.3)	-	3/11 (27.3)	1/6 (16.7)
Oxacillin	45/51 (88.2)	-	13/15 (86.7)	-	0/11	6 (<0.125) *
Cefotaxime	45/51 (88.2)	-	13/15 (86.7)	-	0/11	0/6
Cefuroxime	45/51 (88.2)	-	13/15 (86.7)	-	0/11	0/6
Piperacillin-Tazobactam	45/51 (88.2)	-	13/15 (86.7)	-	0/11	0/6
Imipenem	45/51 (88.2)	-	13/15 (86.7)	-	0/11	0/6
Ampicillin-Sulbactam	41/47 (87.2)	-	12/14 (85.7)	-	0/11	0/6
Meropenem	44/51 (86.3)	-	12/15 (80.0)	-	0/11	0/6
Ciprofloxacin	32/51 (62.7)	-	13/15 (86.7)	-	0/11	6 (0,0,0,1,1,2) *
Amikacin	30/50 (60.0)	-	12/15 (80.0)	-	1/11 (9.1)	4 (0.5,4,8,32) *
Gentamicin	30/51 (58.8)	-	12/15 (80.0)	-	1/11 (9.1)	6 (0,0,0.1,1,2,4) *
Clindamycin	23/51 (45.1)	-	7/15 (46.7)	-	0/11	0/6
Levofloxacin	22/51 (43.1)	12/51 (23.5)	12/15 (80.0)	-	0/11	0/6 ^#^
Moxifloxacin	18/51 (35.3)	5/51 (9.8)	11/15 (73.3)	1/15(6.7)	0/11	0/6
Ceftaroline	6/12 (30.0)	-	3/3 (100)	-	0/8	0/2
Cotrimoxazol	9/50 (18.0)	5/50 (10.0)	11/15 (73.3)	-	0/11	0/6
Fosfomycin	6/51 (11.8)	-	6/15 (40.0)	-	0/11	2/6 (40.0)
Doxycycline	4/50 (8.0)	1/50 (2.0)	1/15 (6.7)	1/15(6.7)	0/11	5/6 (83.3)
Rifampicin	3/51 (5.9)	1/51 (2.0)	2/15 (13.3)	-	0/11	0/6
Daptomycin	0/48	-	1/14 (7.1)	-	1/11 (9.1)	5 (0,0,0,0,0) *
Linezolid	0/51	-	1/15 (6.7)	-	0/11	0/6
Teicoplanin	0/51	-	4/15 (26.7)	-	0/11	0/6
Vancomycin	0/51	-	0/15	-	0/11	0/6

R = resistant, I = susceptible, increased exposure, N = number of tested organisms, * = Minimum inhibitory concentrations (no breakpoint), ^#^ = 2/6 (33.3) of the isolates were susceptible, increased exposure (I) against levofloxacin.

**Table 3 antibiotics-10-00323-t003:** Trends in antibiotic resistance of *Staphylococcus epidermidis* isolates from neonatal sepsis patients, 2013–2020.

Antibiotics	2013–2016 % (N)	2017–2020 % (N)	*p*-Value
Roxithromycin	93.5 (31)	83.0 (20)	0.321
Oxacillin	93.3 (30)	85.0 (20)	0.341
Meropenem	93.1 (29)	85.0 (20)	0.362
PenicillinG	90.3 (31)	90.0 (20)	0.970
Cefotaxime	90.3 (31)	85.0 (20)	0.568
Cefuroxime	90.3 (31)	85.0 (20)	0.568
Piperacillin-Tazobactam	90.3 (31)	85.0 (20)	0.568
Imipenem	90.3 (31)	85.0 (20)	0.568
Ampicillin-Sulbactam	90.3 (31)	81.2 (16)	0.382
Ciprofloxacin	67.7 (31)	70.0 (20)	0.867
Levofloxacin	63.3 (30)	75.0 (20)	0.391
Amikacin	56.7 (30)	65.0 (20)	0.560
Gentamicin	54.8 (31)	65.0 (20)	0.476
Clindamycin	51.6 (31)	35.0 (20)	0.249
Moxifloxacin	40.0 (30)	55.0 (20)	0.302
Cotrimoxazol	26.7 (30)	30.0 (20)	0.799
Fosfomycin	13.3 (30)	10.0 (20)	0.725
Doxycycline	10.0 (30)	10.0 (20)	1.000
Rifampicin	10.0 (30)	5.0 (20)	0.527
Daptomycin	0 (30)	0 (18)	-
Linezolid	0 (31)	0 (20)	-
Teicoplanin	0 (31)	0 (19)	-
Vancomycin	0 (31)	0 (19)	-

% = percentage of resistant (including susceptible, increased exposure) isolates, N = number of tested isolates.

**Table 4 antibiotics-10-00323-t004:** Antibiotic resistance patterns of predominant gram negative bacterial isolates from neonatal sepsis patients, 2012–2020.

Antibiotics	*Escherichia coli* (N = 23)		*Klebsiella oxytoca* (N = 3)		*Klebsiella pneumonia* (N = 2)		*Enterobacter cloacae* (N = 2)	
	R (%)	I (%)	R	I	R	I	R	I
Ampicillin	17 (73.9)	3 (13)	3	-	2	-	2	-
Ampicilin-Sulbactam	12 (52.2)	5 (21.7)	0	-	1	-	2	-
Piperacillin	11 (47.8)	6 (26.1)	1	-	1	-	0	-
Moxifloxacin	6 (26.1)	-	0	-	0	-	0	-
Levofloxacin	5 (21.7)	-	0	-	0	-	0	-
Ciprofloxacin	5 (21.7)	-	0	-	0	-	0	-
Cotrimoxazol	4 (17.4)	-	0	-	0	-	0	-
Ceftibuten	2 (15.4)	-	0	-	NA	-	1	-
Cefuroxime	2 (8.7)	16 (69.6)	0	2	0	2	2	-
Fosfomycin	0	-	2	-	0	-	1	
Cefotaxime	1	-	0	-	0	-	0	-
Ceftazidime	1	-	0	-	0	-	0	-
Aztreonam	1	-	0	-	0	-	0	-
Tobramycin	0	-	0	-	0	-	0	1
Amikacin	0	-	0	-	0	-	0	
Colistin	0	-	0	-	0	-	0	-
Gentamicin	0	-	0	-	0	-	0	-
Imipenem	0	-	0	-	0	-	0	-
Meropenem	0	-	0	-	0	-	0	-
Piperacillin-Tazobactam	0	-	0	-	0	-	0	-

R = resistant, I = susceptible, increased exposure, N= number of tested organisms, NA= not available.

**Table 5 antibiotics-10-00323-t005:** Trends in antibiotic resistance of *Escherichia coli* isolates from neonatal sepsis patients, 2013–2020.

Antibiotics	2013–2016 % (N)	2017–2020 % (N)	*p*-Value
Ampicillin	76.9 (13)	100.0 (10)	0.111
Ampicillin-Sulbactam	69.2 (13)	80.0 (10)	0.569
Piperacillin	69.2 (13)	80.0 (10)	0.569
Cefuroxime	69.2 (13)	90.0 (10)	0.242
Moxifloxacin	38.5 (13)	10.0 (10)	0.132
Levofloxacin	38.5 (13)	0 (10)	0.030
Ciprofloxacin	38.5 (13)	0 (10)	0.030
Cotrimoxazol	23.1 (13)	10.0 (10)	0.422
Cefotaxime	7.7 (13)	0 (10)	0.380
Ceftazidime	7.7 (13)	0 (10)	0.380
Aztreonam	7.7 (13)	0 (10)	0.380
Fosfomycin	0 (13)	0 (10)	-
Doxycycline	0 (13)	0 (10)	
Tobramycin	0 (13)	0 (10)	-
Amikacin	0 (13)	0 (10)	-
Colistin	0 (13)	0 (10)	-
Gentamicin	0 (13)	0 (10)	-
Imipenem	0 (13)	0 (10)	-
Meropenem	0 (13)	0 (10)	-
Piperacillin-Tazobactam	0 (13)	0 (10)	-

% = percentage of resistant (including susceptible, increased exposure) isolates, N = number of tested isolates.

## Data Availability

The data presented in this study are available on request from the corresponding author. All relevant data are within the paper.

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
