# Peer review of "Antibiotic Resistance Patterns of Bacterial Isolates from Neonatal Sepsis Patients at University Hospital of Leipzig, Germany"

_antibiotics, 2021, doi:10.3390/antibiotics10030323_

Round 1

Reviewer 1 Report

Some minor points to be adressed in the paper:

Table 1: Age -Y missing unit (days, years...)

Line125-131: Has tthere been a change of antibiotics usage regime within those years?

Line163-169: Have the isolates been identified to the strain level?Is there data available whether it's the same genotype? Cuod the isolate be traced back to hopsital origin? Was there a change of the bacterial population over the years?

Line 237-238: Why was this decision made? Are there really no alternatives available?

Line 336-341: This is sure a good recomandation and I support it.A monitoring of resistance patterns as well as bacterial population should be done. However, a rotation principle of antibiotics regime may also be an idea (together with monitoring). 

Author Response

Reviewer 1

Comments and Suggestions for Authors

Some minor points to be addressed in the paper:

Response: The reviewer’s points are greatly appreciated. The authors addressed the points raised by the reviewer in the paper.

Comment: Table 1: Age -Y missing unit (days, years...)

Response: Age is corrected as Age in days. Page 3, Table 1, last column.

Comment: Line125-131: Has there been a change of antibiotics usage regime within those years?

Response: Yes, in our hospital, the first-line regimen for empirical treatment of EOS was ampicillin combined with cefotaxime until 2018.  Since 2018 this regimen has been changed into ampicillin combined with gentamicin, and in the absence of clinical improvement or in case of suspicion of meningitis cefotaxime is added. This information is now included in the text on page 9, line 274 -276.

Comments: Line163-169: Have the isolates been identified to the strain level?

Response: No, isolates have been identified only to the species level.

Is there data available whether it's the same genotype?

Response: No, there is no data on the genotype of the isolates.

Cuod the isolate be traced back to hopsital origin?

Response: All isolates were isolated from the same hospital (University of Leipzig hospital).

Was there a change of the bacterial population over the years?

Response: No, there was no change in the bacterial population over the years. Both, S. epidermidies and E. coli remained the predominant gram-positive and gram-negative bacterial populations, respectively over the study period.

Comment: Line 237-238: Why was this decision made? Are there really no alternatives available?

Response: The decision was made in accordant with the recommendation of a recently published review [39]. The use of cefotaxime regularly in first empiric therapy for EOS cases is not advisable. Because previous studies have shown that in first-line therapy it results in more Extended-spectrum beta-lactamase bacteria infections [40] and leads to more invasive fungal infections [41]. Furthermore, Clark and his colleagues [42] found higher mortality in cefotaxime/ampicillin-treated newborns compared with ampicillin/gentamycin-treated newborns. Page 9, Paragraph 3, line 246 – 258.

Comments: Line 336-341: This is sure a good recommendation and I support it. A monitoring of resistance patterns as well as bacterial population should be done. However, a rotation principle of antibiotics regime may also be an idea (together with monitoring). 

Response: As per the reviewer’s suggestion, we have included the following recommendation in the conclusion part together with monitoring. “Moreover, antibiotic rotation through systematically rotating antibiotics or antibiotic classes for empirical treatment might also be helpful to reduce antibiotic resistance”. Page 11, Paragraph 3, Line 360-362.

Reviewer 2 Report

The manuscript written by Belay Tessema et al. is a very interesting manuscript describing the AST patterns from neonatal patients.

This manuscript is well-written but deserve modification to increase its understanding by the reader.

Global : 

Number below twelve must be written in full letters, if appropriate.

Bacterial names have to be italicized

Prefer passive form and not personal experience, as this is not a newspaper article.

Results : 

Has the authors searched for information about the K1 antigens for E. coli responsible for EOS, as this characteristics is well-debated.

Table 1 : precise that age is indicated in days.

Table 2 and 4 : Information must be splitted into to column for 2013-2016 and 2017-2020 periods, as this comparison is crucial for the other part of the manuscript.

Tables : how have the authors considered intermediate AST? as a Resistant strains, I suppose, but please state it.

Results/Discussion : 

AST profiles have to be associated with clinical outcomes, please add this information.

Discussion : 

Limitation : I could not agree with the statement that infection could not be classified between CA- or HA- infection. The delay between hospitalization (or birth) and sampling would answer this question.

Methods : 

How was determined the number of patient to include ? It seems that sometimes, the difference is not present due to a lack of power.

What was the approache used by the authors for the multiple testing correction ?

Author Response

Reviewer 2

Comments and Suggestions for Authors

The manuscript written by Belay Tessema et al. is a very interesting manuscript describing the AST patterns from neonatal patients.

This manuscript is well-written but deserves modification to increase its understanding by the reader.

Response: The reviewer’s comments and suggestions are very much appreciated. The authors responded to the questions and remarks raised by the reviewer. The manuscript has been revised according to the reviewer’s comments and suggestions to increase its understudying by the reader.

 Global : 

Comment: Number below twelve must be written in full letters, if appropriate.

Response:  Corrected. 11 and 5 replaced by full letters. Page 3, Results part, Paragraph1.

Comment: Bacterial names have to be italicized

Response:  Bacterial names have been italicized throughout the manuscript. Example: Page 5, Paragraph 1; Page 6 Paragraph 1; Page7, Paragraph 1.

Comment: Prefer passive form and not personal experience, as this is not a newspaper article.

Response: Corrected throughout the manuscript. Example: page 9, paragraph 3.

Results: 

Comment: Has the authors searched for information about the K1 antigens for E. coli responsible for EOS, as this characteristics is well-debated.

Response: No, unfortunately, we have not searched for information about K1 antigens for E.coli.

Comment: Table 1: precise that age is indicated in days.

Response: Age is corrected as Age in days. Page 3, Table 1, last column.

Comment: Table 2 and 4: Information must be splitted into to column for 2013-2016 and 2017-2020 periods, as this comparison is crucial for the other part of the manuscript.

Response: The same information has already been splitted into two columns for the 2013-2016 and 2017-2020 periods in Tables 3 and 5. The trend analysis results of the most predominant gram-positive and gram-negative bacterial isolates presented in Tables 3 and 5 include the same antibiotics listed in Tables 2 and 4. Moreover, splitting each dependent variable (R and I) into two columns for all four isolates including less frequently isolated organisms will need additional 8 columns for each table. And most of the columns for less frequent isolates will remain without number. Therefore, we would rather maintain the two tables as they are.

Comment: Tables: how have the authors considered intermediate AST? as Resistant strains, I suppose, but please state it.

Response: We have analyzed and presented resistant and intermediate AST results separately in Tables 2 and 4. However, to make a trend analysis in Tables 3 and 5, we had to categorize the AST results only into two categories (S or R). So we had to merge I to either S or R categories. According to the new definition of Intermediate AST result (EUCAST 2019). Intermediate strains cannot be successfully treated with the standard regime dose. That means it needs increased dose of the antibiotics. Therefore, only for the trend analysis tables, we categorized those few isolates with intermediate AST results under the resistant category.

Results/Discussion: 

Comment: AST profiles have to be associated with clinical outcomes, please add this information.

Response: In this study we analyzed retrospective data; unfortunately, we do not have data on the clinical outcomes of the study participants. We have mentioned this information as the limitation of this study. Page 10, paragraph 1.

Discussion: 

Comment: Limitation: I could not agree with the statement that infection could not be classified between CA- or HA- infection. The delay between hospitalization (or birth) and sampling would answer this question.

Response: We agree with the reviewer's statement that infections could be classified as CA or HA based on the duration between neonates’ hospitalization and sampling time. However, in this study, all the data were collected from the laboratory records of the Institute of Medical Microbiology and epidemiology of infectious diseases as it is mentioned in the methods part. In this record, information about the neonates’ hospitalization date is not available. Therefore, we were not able to classify infections as CA or HA.

Methods: 

Comment: How was determined the number of patients to include? It seems that sometimes, the difference is not present due to a lack of power.

Response: As it is mentioned in the methods part, all neonates with blood culture-proven sepsis diagnosed during the study period and with antibiotic susceptibility test results at the University Hospital of Leipzig were included in this study. Page 10, Methods part, paragraph 2

Comment: What was the approache used by the authors for the multiple testing correction ?

Response: Authors used Walker’s method for multiple testing corrections based on the minimum p-value to establish the significance threshold. A p-value of < 0.05 was considered statistically significant. Page 11, Statistical Analysis part, Paragraph 1.

Round 2

Reviewer 2 Report

The authors have answered most of the previsous comments I submitted, Nevertheless, some remains .

Comment: Has the authors searched for information about the K1 antigens for E. coli responsible for EOS, as this characteristics is well-debated.

Response: No, unfortunately, we have not searched for information about K1 antigens for E.coli.

  • Authors have to discuss potential finding associated with this information

Comment: How was determined the number of patients to include? It seems that sometimes, the difference is not present due to a lack of power.

Response: As it is mentioned in the methods part, all neonates with blood culture-proven sepsis diagnosed during the study period and with antibiotic susceptibility test results at the University Hospital of Leipzig were included in this study. Page 10, Methods part, paragraph 2

  • Authors could perform a post hoc detemination of the power of their study, in order to highlight/justify this possible lack of power.

For all the following comments I agree with the author's answer but recommend to state it in the manuscript. Please add these informations.

  • Comment: Tables: how have the authors considered intermediate AST? as Resistant strains, I suppose, but please state it
  • Comment: Limitation: I could not agree with the statement that infection could not be classified between CA- or HA- infection. The delay between hospitalization (or birth) and sampling would answer this question.

Author Response

Comments and Suggestions for Authors

The authors have answered most of the previous comments I submitted, Nevertheless, some remains.

Response: The authors very much appreciate the reviewer`s thorough evaluation of our work for the second time. The comments and suggestions have been used to improve the article further.

Comment: Has the authors searched for information about the K1 antigens for E. coli responsible for EOS, as this characteristics is well-debated. Authors have to discuss potential finding associated with this information.

Response: We have discussed K1 antigen for E. coli in the discussion part. Page 8, paragraph 1.

Comment: How was determined the number of patients to include? It seems that sometimes, the difference is not present due to a lack of power. Authors could perform a post hoc detemination of the power of their study, in order to highlight/justify this possible lack of power.

Response: Power analysis is an indispensable component of planning clinical research studies. However, when used to indicate power for outcomes already observed (post hoc power or retrospective power), it is not only conceptually flawed but also analytically misleading. Recently published simulation results clearly showed that such power analyses do not indicate true power for detecting statistical significance, since post hoc power estimates are generally variable in the range of practical interest and can be very different from the true power (https://gpsych.bmj.com/content/gpsych/32/4/e100069.full.pdf). Other authors also documented that post hoc power analysis is not useful (https://dirnagl.files.wordpress.com/2014/07/hoenig-heisey-the-abuse-of-power2001.pdf). Similarly, many of the experts in the area are advising to avoid doing post hoc power analysis. Therefore, we refrained from adding such arguable information in our paper to not affect the quality of our paper.

For all the following comments I agree with the author's answer but recommend to state it in the manuscript. Please add these informations.

Comment: Tables: how have the authors considered intermediate AST? as Resistant strains, I suppose, but please state it

Response:  This information is added in the method part. Page 11, paragraph 1.

Comment: Limitation: I could not agree with the statement that infection could not be classified between CA- or HA- infection. The delay between hospitalization (or birth) and sampling would answer this question.

Response: This information is added in the limitation part. Page 9, paragraph 5.